# Acetate and Butyrate Improve β-cell Metabolism and Mitochondrial Respiration under Oxidative Stress

**DOI:** 10.3390/ijms21041542

**Published:** 2020-02-24

**Authors:** Shuxian Hu, Rei Kuwabara, Bart J. de Haan, Alexandra M. Smink, Paul de Vos

**Affiliations:** Department of Pathology and Medical Biology, University Medical Center Groningen, University of Groningen, Hanzeplein 1, EA11, 9713 GZ Groningen, The Netherlands; r.kuwabara@umcg.nl (R.K.); b.j.de.haan@umcg.nl (B.J.d.H.); a.m.smink@umcg.nl (A.M.S.); p.de.vos@umcg.nl (P.d.V.)

**Keywords:** acetate, butyrate, pancreatic β-cells, apoptosis, mitochondrial dynamics, oxygen consumption rate

## Abstract

Islet dysfunction mediated by oxidative and mitochondrial stress contributes to the development of type 1 and 2 diabetes. Acetate and butyrate, produced by gut microbiota via fermentation, have been shown to protect against oxidative and mitochondrial stress in many cell types, but their effect on pancreatic β-cell metabolism has not been studied. Here, human islets and the mouse insulinoma cell line MIN6 were pre-incubated with 1, 2, and 4 mM of acetate or butyrate with and without exposure to the apoptosis inducer and metabolic stressor streptozotocin (STZ). Both short-chain fatty acids (SCFAs) enhanced the viability of islets and β-cells, but the beneficial effects were more pronounced in the presence of STZ. Both SCFAs prevented STZ-induced cell apoptosis, viability reduction, mitochondrial dysfunction, and the overproduction of reactive oxygen species (ROS) and nitric oxide (NO) at a concentration of 1 mM but not at higher concentrations. These rescue effects of SCFAs were accompanied by preventing reduction of the mitochondrial fusion genes *MFN*, *MFN2*, and *OPA1*. In addition, elevation of the fission genes *DRP1* and *FIS1* during STZ exposure was prevented. Acetate showed more efficiency in enhancing metabolism and inhibiting ROS, while butyrate had less effect but was stronger in inhibiting the SCFA receptor GPR41 and NO generation. Our data suggest that SCFAs play an essential role in supporting β-cell metabolism and promoting survival under stressful conditions. It therewith provides a novel mechanism by which enhanced dietary fiber intake contributes to the reduction of Western diseases such as diabetes.

## 1. Introduction

Acetate and butyrate are short-chain fatty acids (SCFAs) and the main large intestinal microbiota fermentation products of soluble fibers. SCFAs have been reported to attenuate various inflammatory responses and metabolism disorders [1,2]. For example, acetate and butyrate have been shown to reduce or delay the development of autoimmune diabetes by inhibiting proinflammatory lymphocytes and supporting regulatory lymphocytes [3,4]. Furthermore, SCFAs are reported to promote insulin sensitivity and metabolism in obese individuals by increasing fat oxidation, energy expenditure, and decreasing lipolysis [5,6,7,8]. The effects of SCFA are not restricted to the intestine, as they have been reported to enter the peripheral circulation and to participate in peripheral tissue metabolism. In the systemic circulation, they affect energy homeostasis and metabolism by regulating mitochondrial functions and dynamics in brown adipocytes, liver, and skeletal muscle via G protein-coupled receptor (GPR) 41 and 43 [9,10,11,12,13]. Although glucose metabolism has shown to be impacted by acetate and butyrate, there are no studies yet that address the effects of SCFAs on pancreatic islets [14,15]. 

Islets are known to express the SCFA receptors GPR 41 and 43, indicating that SCFAs might be involved in islet-cell metabolism and mitochondrial function [16,17]. Mitochondrial dysfunction of the insulin-producing β-cells might lead to β-cell apoptosis through the generation of free radicals and eventually to metabolic disorder and diabetes [18,19]. SCFAs might beneficially impact several processes in the mitochondria of β-cells. They might affect mitochondrial fusion proteins, e.g., atrophy (*OPA1*) and mitofusin (*MFN*) 1 and 2, and fission proteins, e.g., dynamin-related protein 1 (*DRP1*) and fission protein 1 (*FIS1*) [20]. An increase in fusion proteins might protect cells against toxin-induced apoptosis, whereas fission is associated with the generation of mitochondrial fragments, which occurs in the early stages of cell apoptosis [21]. If SCFAs impact these proteins, it might be involved in the support of mitochondrial function, efficiency, and dynamics during stress, which might contribute to the prevention of β-cell dysfunction and save β-cells both in type 1 and type 2 diabetes.

To gain insight in a possible role of acetate and butyrate in β-cell function, we investigated the impact of acetate and butyrate on the viability and mitochondrial function of both human islets and a mouse β-cell line, as no adequate human β-cell line was available. This was done in the presence and absence of the well-known apoptosis inducer in β-cells, streptozotocin (STZ). The impact and rescuing effects of acetate and butyrate on basal respiration, ATP-linked respiration, and spare respiration was investigated. As STZ induces cell death via the generation of free radicals and nitric oxide (NO), the impact of acetate and butyrate was also studied on the generation of these molecules during exposure to STZ. Our data demonstrate that SCFAs enhance the metabolism of β-cells and can protect cells from many deleterious processes in an SCFA-dependent fashion.

## 2. Results

### 2.1. SCFAs Influence Human Islet Cell Viability in a Dose-Dependent Manner 

To investigate the potential role of SCFAs in the support of islet viability, we tested the impact of acetate and butyrate on human cadaveric pancreatic islets. As shown in Figure 1A, acetate had a beneficial effect on islet viability, as it induced a 20.5 ± 4.4% (*p* < 0.01) increase at 1 mM compared to the untreated control cells. However, the higher dose of 4 mM acetate significantly decreased islet viability by 20.3 ± 3.3% (*p* < 0.001). Butyrate also had a dose-dependent effect on islet viability; 1 mM butyrate showed a significant increase of 25.7 ± 3.2% (*p* < 0.001) compared to untreated controls, whereas the higher dosage of 4 mM led to a significant decrease in cell viability by 13.2 ± 4.1% (*p* < 0.05).

Since acetate and butyrate at a concentration of 1 mM positively influenced islet viability, we performed immunofluorescent staining on human islets to confirm the expression of the two major receptors for SCFAs GPR41 and 43. As shown in Figure 1B,C, both GPR41 and 43 are expressed in human islets. The expression of GPR41 was reduced when the islets were incubated with SCFAs. Acetate at high concentration, i.e., 2 mM and 4 mM, decreased the GRP41 expression, and butyrate at 1, 2, and 4 mM inhibited expression of this receptor. Neither of the tested SCFAs influenced the expression of GPR43.

To investigate the impact of the tested SCFAs on insulin synthesis, islets were stained with insulin antibody after incubation with acetate or butyrate at a concentration of 1, 2, and 4 mM for 24 h. As shown in Figure 1D, neither of the tested SCFAs influenced the expression of insulin.

### 2.2. SCFAs Influence β-cell Viability

Islets are clusters of several different cell types. To investigate the involvement of the insulin-producing β-cells in the SCFAs-induced effect, we determined the effect of our SCFAs on a mouse MIN6 β-cell line. Acetate and butyrate also had a dose-dependent effect on β-cell viability (Figure 2A). Acetate at 1 mM showed a significant increased viability of 22 ± 5.9% (*p* < 0.05) compared to untreated controls, whereas acetate at 4 mM a decrease in viability with 27.6 ± 6.2% (*p* < 0.01) was observed. Butyrate at 1 mM also had a beneficial effect on β-cell viability, as it induced a 29.6 ± 5.0% (*p* < 0.001) increase in cell viability compared with the untreated control. Butyrate at higher concentrations, i.e., 2 mM and 4 mM, did not influence viability.

To further confirm the presence of SCFA receptors on β-cells, PCR and Western blot were performed on RNA (Appendix A) and protein isolated from MIN6 cells (Figure 2B). Neither GPR41 nor GPR43 mRNA changed before or after incubation with the tested SCFAs. However, the protein expression of GPR41 was suppressed by 2 and 4 mM acetate, resulting in a significant decline by 19.2 ± 4.3% (*p* < 0.05) and 32.0 ± 9.6% (*p* < 0.01), respectively (Figure 2C). There was no decline at a concentration of 1 mM butyrate, while at 1, 2, and 4 mM, butyrate significantly suppressed the expression of GPR41 by respectively 63.8 ± 2.3%, 64.9 ± 6.2% and 59.9 ± 4.0% (*p* < 0.001). The tested SCFAs did not influence the protein expression of GPR43 (Figure 2D).

### 2.3. SCFAs Prevent Islet Cell-Death During Exposure to STZ

To investigate the effects of acetate and butyrate on human islets under stress, islet viability was also quantified before and after exposure to the β-cell toxin STZ. 

STZ markedly decreased the islet cell viability in human islets by 51.4 ± 6.5% (*p* < 0.001) (Figure 3A). However, pre-incubation with 1 mM acetate prevented STZ-induced cell death with 45.4 ± 9.9% (*p* < 0.01). In addition, 1 mM butyrate prevented STZ-induced cell death, as 45.6 ± 9.1% (*p* < 0.01) more cells stayed viable after STZ exposure. Higher concentrations were tested as well, but rescuing effects were only observed at 1 mM SCFAs.

STZ not only decreased islet cell metabolism, it also significantly induced islet-cell apoptosis in 64.2 ± 5.4% (*p* < 0.001) (Figure 3B,C) of the cells. This effect was suppressed by both acetate and butyrate at 1 mM. Acetate exposure at 1 mM inhibited apoptosis by 53.1 ± 5.5% (*p* < 0.001) and butyrate at the same concentration ameliorated apoptosis with 53.4 ± 5.4% (*p* < 0.001). At concentrations of 2 and 4 mM, acetate could not prevent the STZ-induced apoptosis. Butyrate at 2 mM effectively ameliorated excessive apoptosis, but at a concentration of 4 mM SCFAs, it could not prevent STZ-induced apoptosis.

### 2.4. SCFAs Promote β-Cell Survival in the Presence of STZ 

To determine whether the tested SCFAs influence β-cell survival under stress, cell viability and apoptosis were also investigated in STZ-exposed MIN6 cells. In concordance with the results with human islets, STZ markedly decreased MIN6 β-cell viability by 47.0 ± 4.7% (*p* < 0.001) (Figure 3D). However, a pre-incubation with low-dosage SCFAs prevented this decrease. Acetate in a concentration of 1 mM prevented STZ-induced cell apoptosis with 32.9 ± 11.3% (*p* < 0.01). In addition, 1 mM butyrate prevented apoptosis with 31.5 ± 3.9% (*p* < 0.001). Higher concentrations were tested, but also here, the rescuing effects were only observed at 1 mM SCFAs.

STZ also induced apoptosis in 32.5 ± 1.3% of the β-cells. This pro-apoptotic effect was significantly suppressed by both acetate and butyrate in a dose-dependent manner in STZ-treated MIN6 cells (Figure 3E). Acetate exposure at a concentration of 1 and 2 mM markedly prevented apoptosis with 20.6 ± 1.7% (*p* < 0.001) and 5.8 ± 2.3% (*p* < 0.05). At 4 mM, acetate could not prevent the STZ-induced apoptosis increase. Butyrate at 1 and 2 mM significantly prevented apoptosis with 21.8 ± 2.0% (*p* < 0.001) and 7.7 ± 1.7% (*p* < 0.01). In addition, butyrate at a concentration of 4 mM could not ameliorate the apoptosis induced by STZ.

### 2.5. SCFAs Support β-cell Mitochondrial Respiration and Protect Against STZ-induced β-cell OCR Reduction

To determine how acetate and butyrate beneficially impact cell metabolism, we determined the oxygen consumption rate (OCR), which is an indicator of mitochondrial respiration in β-cells (Figure 4A) [22]. We used OCR and not glucose-induced insulin secretion, as it has been reported that OCR correlates better and allows for the quantification of smaller differences in viability and function [23]. It was only done at 1 mM SCFAs, as we found beneficial effects at this concentration. As these assays require a high amounts of cells, it could only be performed with MIN6 cells and not with the rarely available human islets. As shown in Figure 4B–E, acetate induced a statistically significant beneficial effect on β-cell OCR and stimulated all mitochondrial respiration processes. Acetate enhanced basal respiration by 22.2 ± 7.6% (*p <* 0.05) (Figure 4C), ATP-linked respiration by 23.7 ± 7.7% (*p* < 0.05) (Figure 4D), and spare respiration by 24.4 ± 6.2% (*p* < 0.01) (Figure 4E). Butyrate did not boost β-cell mitochondrial respiration compared to the control group.

To investigate whether the tested SCFAs also protect β-cell respiration under STZ-induced stress, MIN6 cell OCR was quantified before and after exposure to STZ with and without exposure to SCFA. STZ has a strong negative impact on OCR (Figure 4B), and it was differentially prevented by acetate and butyrate in β-cells. MIN6 cells incubated with STZ showed a decreased respiration rate, including a significantly decreased basal respiration by 44.0 ± 5.3% (*p* < 0.001), reduced ATP production by 63.0 ± 7.8% (*p* < 0.001), and spare respiration by 58.3 ± 4.7% (*p* < 0.001) (Figure 4B–E). Acetate almost completely prevented these negative effects of STZ. Mitochondrial basal respiration rate, spare respiration, and ATP-linked respiration reduction stayed identical to the untreated controls when cells were exposed to acetate and STZ. Butyrate had a partial rescuing effect. Cells exposed to butyrate before STZ treatment had a 37.7 ± 2.4% (*p* < 0.01) higher basal respiration, 53.8 ± 5.6% (*p* < 0.01) higher ATP-linked respiration, and 46.5 ± 3.5% (*p* < 0.001) higher spare respiration than cells exposed to STZ without butyrate.

### 2.6. SCFAs Affect β-cells Mitochondrial Dynamics under STZ-Induced Stress

In order to gain insight into the possible mechanisms involved in the rescuing effects of acetate and butyrate on mitochondrial respiratory functions after the STZ exposure of β-cells, we studied the effect of acetate and butyrate on mitochondrial fission and fusion, as these are important processes for functional mitochondria after cell damage [12]. Islets (Figure 5A) or MIN6 cells (Figure 5B) were first incubated with acetate or butyrate in the absence of STZ to investigate the influence of the tested SCFAs on healthy insulin-producing β cells. Under normal culture conditions, acetate and butyrate did not affect the expression of mitochondrial dynamic genes. 

However, this was different following STZ exposure. After incubation with STZ, the fusion genes *MFN1*, *MFN2*, and *OPA1* were found to be statistically significantly downregulated in human islets by respectively 65.3 ± 14.0% (*p <* 0.01), 75.5 ± 12.5% (*p* < 0.001), and 57.1 ± 11.8% (*p* < 0.01) (Figure 5C). Furthermore, the expression of the fission genes *DRP1* and *FIS1* was statistically significantly increased in STZ-treated human islets by respectively 68.4 ± 15.5% (*p* < 0.01) and 66.0 ± 12.4% (*p* < 0.001). This indicates that STZ induces a reduction in fusion and a stimulation of fission in human islets. Under STZ-induced mitochondrial stress, acetate prevented against the STZ-induced reduction of the fusion genes *MFN1* (*p* < 0.001) and *MFN2* (*p* < 0.001), which were comparable with the untreated control groups. The reduction of *OPA1* was prevented by 45.3 ± 12.9% (*p* < 0.05). Additionally, acetate prevented the increase in the fission genes *DRP1* with 65.3 ± 5.9% (*p* < 0.01) and *FIS1* with 55.5 ± 7.1% (*p* < 0.05). In addition, butyrate showed similar pro-fusion and anti-fission effects. Butyrate almost completely prevented the STZ-induced decrease in the fusion genes *MFN1* (*p* < 0.01) and *MFN2* (*p* < 0.001) compared to islets exposed to only STZ. The decrease of another mitochondrial fusion gene *OPA1* was prevented by 47.1 ± 17.6% (*p* < 0.05) compared to islets with only STZ-exposure. In addition, butyrate moderated the upregulation of fission gene *DRP1* (*p* < 0.01) to control level and *FIS1* by 47.1 ± 17.6% (*p* < 0.01).

We repeated the experiments with MIN6 β-cells. In addition, here, the presence of STZ showed a pro-fission and anti-fusion effect. The fusion genes *MFN1*, *MFN2*, and *OPA1* were found to be statistical significantly downregulated in β-cells by respectively 23.4 ± 2.7% (*p* < 0.01), 36.2 ± 5.4% (*p* < 0.001), and 56.9 ± 2.3% (*p* < 0.001) (Figure 5D). The expression of the fission genes *DRP1* and *FIS1* were upregulated by 48.1 ± 4.3% (*p* < 0.001) and 43.9 ± 4.0% (*p* < 0.001). Both acetate and butyrate showed pro-fusion and anti-fission effects in STZ-treated MIN6 cells, which prevented the decrease in the fusion genes *MFN1* (*p* < 0.001), *MFN2* (*p* < 0.001), and *OPA1* (*p* < 0.001); it also prevented the increase in the fission genes *DRP1* (*p* < 0.001) and *FIS1* (*p* < 0.001) compared to cells with only STZ-exposure.

### 2.7. SCFAs Attenuate STZ-induced Generation of Free Radicals 

As ROS and NO release are hallmarks of STZ-induced islet toxicity, we also studied whether acetate and butyrate impact ROS and NO release in islets [24]. In human islets, STZ increased the production of ROS by 59.3 ± 3.6% (*p* < 0.001) (Figure 6A). This increase was prevented by acetate (*p* < 0.001) as well as by butyrate (*p* < 0.001). Notably, acetate showed a significantly more effective anti-ROS (reactive oxygen species) effect in human islets (*p* < 0.05). STZ had a strong impact on NO synthesis in human islets, which was enhanced by 68.6 ± 4.2% (*p* < 0.001) (Figure 6B). Acetate prevented nearly half of this STZ-induced NO production (*p* < 0.001). Butyrate more effectively inhibited the STZ-induced NO production (*p* < 0.001), whose efficiency was 39.0 ± 5.1% (*p* < 0.001) higher than that of acetate (Figure 6B). 

We also studied ROS and NO generation induced by STZ in a MIN6 cell line. As shown in Figure 6C, after incubation with STZ, ROS generation was statistically significantly increased with 55.1 ± 4.7% (*p* < 0.001) in the STZ-treated MIN6 cells. Acetate inhibited this STZ-induced increase (*p* < 0.001), as well as butyrate (*p* < 0.001). STZ enhanced NO in MIN6 cells with 36.8 ± 3.5% (*p* < 0.001) (Figure 6D). Acetate prevented this STZ-induced increase (*p* < 0.001) as well as butyrate (*p* < 0.001). In the MIN6 cell line, there was no significant difference in the ability of acetate and butyrate to reduce free radical generation. Our findings suggest that the tested SCFAs inhibit STZ-induced oxidative and nitrosative stress in islet β-cells by reducing ROS and NO levels, but the efficiency in human islets is SCFA-dependent.

## 3. Discussion

Pancreatic β-cells are very sensitive to oxidative stress-induced mitochondrial dysfunction due to a relatively weak enzymatic anti-oxidative defense mechanism along with an inefficient repair system for oxidative DNA damage [25,26]. This is a major cause of islet loss in the process toward both type 1 and type 2 diabetes. Many report the beneficial effects of SCFA on glucose metabolism [14,15], but most of these effects are attributed to the increasing insulin sensitivity of peripheral organs rather than directly impacting β-cells to keep producing insulin under stress. Previous studies have shown that SCFAs may stimulate the antioxidant defense against e.g., ischemia–reperfusion injury in cell types other than β-cells such as in brain cells [27]. In addition, SCFAs have been shown to support mitochondrial function in mouse liver, skeletal muscle, and fat cells [12,13,28]. This was the main rational to test the impact of SCFAs on β-cells [29,30]. In addition, we tested how SCFAs can reduce oxidative and nitrosative stress in human islets and β-cells after exposure to STZ [31]. We demonstrate the strong viability promoting effects and protection against oxidative and nitrosative stress by both acetate and butyrate at lower concentrations of 1 mM, while higher concentrations had adverse effects. 

Both human islets and a β-cell line were studied. Despite amble experience with isolating β-cells from human islets [32], we were unable to collect sufficient cells for reliably performing OCR on the cells. The applied OCR technique requires too many islets. Therefore, we choose the strategy to use MIN6 cells for the assays where high numbers of islets are needed. Here, with the usage of both human islets and the MIN6 mouse β-cell line, we proved the direct effect of SCFAs on islets and β-cells. In human islets, we show that acetate and butyrate at 1 mM supported cell metabolism and prevented cell death. In the presence of STZ, the impact of both SCFAs was even more pronounced. Several studies have demonstrated that STZ induces damage mainly by provoking ROS and NO production in islet cells [33,34]. Therefore, as a possible mechanism for the protective effects of SCFAs, we measured after exposure to SCFAs the ROS/NO levels, mitochondrial function, and expression of GPR receptors. Incubation with 1 mM of both SCFAs strongly attenuated STZ-induced overproduction of ROS and NO, which might be caused by multiple mechanisms, including the amelioration of mitochondrial damage by supporting mitochondrial metabolism as an energy source. These two SCFAs seem to be able to reduce this oxidative stress, leading to improved islet-cell survival via, as we show here, regulating essential genes involved in mitochondrial dynamics. We observed a shift toward the fusion genes (*MFN1*, *MFN2*, and *OPA1*) and a downregulation of the mitochondrial fission genes (*DRP1* and *FIS1*) [12,35]. The shift toward mitochondrial fusion is known to improve not only mitochondrial cell energy metabolism but also to reduce radical overproduction [36,37].

The SCFAs effects were dose-dependent; acetate and butyrate at a concentration of 1 mM enhanced β-cell viability and showed slightly improved metabolism at 2 mM, but no protective effects were observed at 4 mM. The physiological concentrations of acetate and butyrate in the portal and peripheral circulation were also found to be approximately 1 mM [38,39]. The loss of protection at higher concentrations and observed enhanced apoptosis at 4 mM with both SCFAs might be caused by multiple mechanisms. A plausible explanation is a lowering of intracellular pH induced by excessive SCFAs entering β-cells via free diffusion. Other explanations are that the oxidative and mitochondrial stress caused by an incomplete fatty acid oxidation increases ketone bodies in the cell and causes cell death [40,41]. Moreover, a higher concentration of SCFAs may upregulate the transcription of cellular growth inhibitory genes by modifying their histone acetylation status, leading to cytotoxicity [42,43].

Acetate had a different effect than butyrate. Acetate showed more efficacy in supporting energy metabolism and anti-oxidative effects than butyrate. This might be caused by adaptations of β-cells to lower concentrations of butyrate in the pancreatic circulation. Butyrate is present at relatively lower concentrations compared to acetate because of the preferential use of butyrate by colonic epithelial cells for energy production [44]. Although there are no studies accurately quantifying and reporting the physiological concentration of acetate and butyrate surrounding islets, it is likely that butyrate concentrations will be lower and that β-cells will already benefit from butyrate at a lower concentration than that of acetate. However, we noticed that butyrate showed stronger effects on inhibiting GPR41 and NO generation, which suggests that butyrate function in β-cells might rely on a signal-mediated process via the SCFA receptor GPR41. Butyrate is sensed by GPR41 and stimulates a cascade of the signal transduction pathway [45,46], resulting in faster and more efficient energy metabolism toward β-oxidation, and the suppressed synthesis of inducible nitric oxide synthase.

Recently, acetate and propionate have been reported to improve insulin secretion and to protect against apoptosis in β-cells damaged by the cytokines interferon gamma, tumor necrosis factor alpha, and interleukin 1 beta and the fatty acid sodium palmitate [16,29]. Pingitore et al. demonstrate that acetate and propionate stimulate insulin secretion via the phospholipase C/protein kinase C pathways and that effects are dependent on GPR43. The anti-apoptotic impact of acetate and propionate are also reported to be GPR43-dependent [16]. Here, we investigate the effect of acetate but also of another important SCFA butyrate in rescuing β-cells compromised by an insulin-producing β-cell specific toxin, STZ, that disturbs mitochondrial metabolism [24]. We demonstrate that both acetate and butyrate rescue STZ-impaired β-cells mitochondrial respiration and promote mitochondrial repair by inducing a shift toward the fusion process in mitochondria. This is, to the best of our knowledge, a new mechanism by which SCFAs contribute to rescue of damaged β-cells. Butyrate is an important SCFA and was not studied by Pingitore et al., but it was shown in our study to contribute together with acetate to rescue β-cell mitochondrial processes such as respiration, ATP generation, and dynamics when exposed to STZ. This underlines and corroborates previous studies [16,29], demonstrating that SCFAs, which are microbial fermentation products of dietary fibers, can play a role in the prevention, delaying, or treatment of diabetes by making β-cells more resistant to oxidative stress. This is, to the best of our knowledge, a new mechanism by which dietary fiber intake can contribute to lowering of the long list of Western diseases. The lower intake of dietary fiber by consumers in Western countries compared to more traditional societies has been suggested to be one of the major causes of the increasing occurrence of both type 1 and 2 diabetes [47]. Previous studies explained the effects of dietary fiber intake and the lowering of diabetes in consumers as an attenuating effect on inflammatory cells, inhibition of absorption of glucose and fat in the intestine, and increased insulin sensitivity [48,49]. Our study demonstrates and suggests that SCFAs formed from dietary fibers also play an essential role in supporting β-cell metabolism and promoting survival under stressful conditions. 

In conclusion, this study provides new insights into how SCFAs can contribute to the maintenance of health. It shows that acetate and butyrate support β-cell metabolism, modulate mitochondrial efficiency and dynamics, and by that provide islet cells with enhanced anti-oxidative responses by which less ROS and NO is generated when exposed to the β-cell specific toxin STZ. Therefore, they support pancreatic β-cells function and prevent STZ-induced cell apoptosis.

## 4. Materials and Methods 

### 4.1. Cell Culture

The MIN6 cell line was purchased from American Type Culture Collection (ATCC, Manassas, VA, USA). MIN6 cells (passages 30–45) were cultured in (Dulbecco’s Modified Eagle Medium) DMEM High-glucose medium (Lonza, Basal, Switzerland), containing 15% fetal bovine serum (FBS, Lonza), 50 µmol/L β-mercaptoethanol, 2 mmol/L l-glutamine, 50 U/mL penicillin, and 50 µg/mL streptomycin (all from Sigma-Aldrich, St. Louis, MO, USA). Cells were cultured at 37 °C in a humidified atmosphere containing 95% air and 5% CO_2_.

### 4.2. Human Islet Isolation 

Human islets of Langerhans were isolated from cadaveric pancreata at the Leiden University Medical Center, as previously described [50] and provided through the JDRF award 31-2008-416 (European Consortium for Islet Transplantation, Islet for Basic Research program, Milan, Italy). The detailed information of islet and donor data are mentioned in Appendix A. Islets were used for research when the quality and/or number were insufficient for clinical application according to national laws and when research consent was available. All the procedures were approved and carried out in accordance with the code of proper secondary use of human tissue in The Netherlands as formulated by the Dutch Federation of Medical Scientific Societies. After shipment to the University Medical Center Groningen, islets were cultured in CMRL-1066 (GIBCO, Bleiswijk, the Netherlands), supplemented with 10% FBS, 50 U/mL penicillin, and 50 µg/mL streptomycin, as previously described [51]. 

### 4.3. Cell Viability

The effect of SCFAs on β-cell viability was determined by the cell proliferation reagent WST-1 (Roche, Indianapolis, IN, USA). Briefly, human islets (15 islets/well) and MIN6 cells (1 × 104 cells/well) were seeded in 96-well plates. Islets and cells were cultured overnight and the following day incubated with or without the acetate (Sigma-Aldrich) or butyrate (Sigma-Aldrich) at 1, 2, and 4 mM for 24 h followed by WST-1 assay. In order to investigate the protective effect of tested SCFAs, a one-hour preincubation of SCFA was followed by a 24-h co-incubation of the SCFAs and 5 mM STZ (Sigma-Aldrich), since STZ is widely reported as causing β-cell destruction and reduced insulin secretion [52]. Cells without SCFAs and STZ incubation were used as negative control, and cells with only STZ incubation were used as positive control. After 30 min incubation with WST-1 (10 µL/well) at 37 °C, the absorbance was measured at 450 nm using a Bio-Rad Benchmark Plus microplate spectrophotometer reader (Bio-Rad Laboratories B.V, Veenendaal, The Netherlands).

### 4.4. Cell Apoptosis 

The effect of acetate and butyrate on β-cell apoptosis was analyzed using an Annexin V (Biolegend, San Diego, CA, USA) and PI (PI, Thermo Scientific, Eugene, OR, USA) staining. Briefly, human islets were seeded on gelatin-coated coverslips with a diameter of 10.2 mm (15 islets/slide). MIN6 cells (1 × 106 cells/well) were cultured in 6-well plates. Islets and MIN6 cells were incubated with or without acetate or butyrate at 1, 2, and 4 mM for 24 h followed by Annexin V/propidium iodide (PI) staining. In order to investigate the rescue effects of SCFAs, one-hour pre-incubation of tested SCFA was followed by a 24-h co-incubation of the SCFAs and 5 mM STZ. Cells without tested SCFAs and STZ incubation were used as negative control. Cells with only STZ incubation were used as positive control. After incubation, islets and MIN6 cells were incubated for 15 min with 4.5 µg/mL Alexa Fluor^®^ 488 Annexin V and 1 µg/mL propidium iodide. The staining of human islets was analyzed using a Leica DM4000 B microscope (Leica Microsystems, Wetzlar, Germany). The percentage of Annexin V positive area and islet size were measured using ImageJ software (v. 1.47; National Institutes of Health, Bethesda, MD, USA). MIN6 cells were analyzed by flow cytometry (BD Biosciences, Breda, The Netherlands). The flow cytometric results were analyzed using FlowJo^®^ 10.4.2 software (LLC, Ashland, USA), and the number of apoptotic cells was expressed as the percentage of the total number of cells. 

### 4.5. Mitochondrial Function 

The effect of acetate and butyrate on mitochondrial function was measured by seeding MIN6 cells in XFe24 cell culture microplates (Seahorse Bioscience, North Billerica, MA, USA) at 4 × 104 cells/well. Cells were cultured as described above. Cells with STZ incubation were used as positive controls, and cells without SCFAs and STZ incubation were used as negative controls. The cells were washed with extracellular flux (XF) basic medium (Seahorse Bioscience) supplemented with 3 mM glucose, 2 mM glutamine, and 2 mM sodium pyruvate followed by an incubation in this medium for 60 min at 37 °C in a non-CO_2_ incubator. Plates were transferred to a Seahorse Bioscience XFe24 extracellular flux analyzer (Seahorse Bioscience) and subjected to an equilibration period. After measuring the basal oxygen consumption rate (OCR) for three cycles, oligomycin (1 µM, Sigma-Aldrich) was added to determine the proportion of respiration used to generate ATP. Then, carbonyl cyanide-4-(trifluoro methoxy) phenyl hydrazone (FCCP, 1 µM, Sigma-Aldrich) was added to determine the maximal respiration by mitochondria. After that, rotenone (0.5 µM, Sigma-Aldrich) and antimycin A (0.5 µM, Sigma-Aldrich) were added to measure the non-mitochondrial respiratory rate.

### 4.6. Reactive Oxygen Species (ROS) Assay

Intracellular ROS was detected using a DCFDA Cellular ROS Detection Assay Kit (Abcam, Cambridge, UK) according to the instructions of the kit manufacturer. Briefly, MIN6 cells (1 × 104/well) and human islets (10 islets/well) were seeded in black 96-well immunoplates (Thermo Scientific). Both were cultured and incubated with the tested SCFAs and STZ as described above. They were incubated with 20 µM 2′,7′-dichlorofluorescin diacetate (DCFDA) at 37 °C for 30 min, and fluorescence was measured (excitation 485 nm, emission 535 nm) using a fluorescence plate reader (PerkinElmer, Waltham, MA, USA).

### 4.7. Nitric Oxide 

The NO concentration in the supernatant of the MIN6 cells and the human islets was measured with a Nitric Oxide Assay Kit (Invitrogen, Vienna, Austria) according to the manufacturer’s instructions. MIN6 cells and human islets were cultured and incubated with the tested SCFAs and STZ as described above. Briefly, nitrate in the supernatants was converted to nitrite using nitrate reductase. Subsequently, using Griess reagents, the nitrite was converted to a deep purple azo compound. Then, a Bio-Rad Benchmark Plus microplate spectrophotometer reader at 540 nm was used to measure the level of azo compounds, which reflected the NO amount in the sample.

### 4.8. Immunofluorescence

Immunofluorescent staining was performed to stain for the SCFA receptors GPR41 and GRP43 and insulin. Human islets were cultured and incubated with the tested SCFAs and STZ as described above. After that, islets were fixed for 15 min with 4% paraformaldehyde (Merck, Darmstadt, Germany), and non-specific binding was blocked by incubating with 1% bovine serum albumin (Sigma-Aldrich) for 1 h. The cells and islets were incubated for 1 h at room temperature with the primary antibodies rabbit anti-GPR41 (Thermo Scientific, 1:100), rabbit anti-GRP43 (Merck, 1:100), and guinea pig anti-insulin (Dako, Santa Clara, CA, USA, 1:200). After washing, the cells and islets were incubated for 1 h at room temperature with donkey-anti-rabbit Alexa Fluor^®^ 488 (Thermo Scientific, 1:500) or goat-anti-guinea pig Alexa Fluor^®^ 488 (Thermo Scientific, 1:500) followed by a 1-min incubation with (4′,6-Diamidino-2-Phenylindole) DAPI (1.0 µg/mL; Roche). The staining of GPR41 and 43 was analyzed using a Leica DM4000 B microscope through the hanging drop method [53]. To this end, islets were placed on a cover slip in a 10 µL phosphate-buffered saline (PBS) micro-spheroid and subsequently placed upside down, allowing scanning of the islets throughout the islets. Insulin staining was analyzed using a Leica SP8 confocal microscope. Data were processed using ImageJ Software.

### 4.9. Western Blot

Western blot was performed to analyze the protein expression of the SCFA receptors GPR41 and GPR43. MIN6 cells were cultured as described above and were lysed afterwards using radioimmunoprecipitation assay buffer (10×, Merck) supplemented with protease inhibitor (Roche). The protein concentration was determined using the BCA™ Protein Assay Kit (Thermo Scientific). Proteins were separated by loading a 10% polyacrylamide gel with 20 µg protein per well. The electrophoresis-separated proteins were subsequently transferred to Immobilon^®^-FL polyvinylidene fluoride membranes (Merck) and blocked with Odyssey^®^ blocking buffer (LI-COR, Leusden, The Netherlands) for 1 h at room temperature. Then, membranes were incubated with the primary antibody rabbit anti-GPR41 (Thermo Scientific, 1:1000) and rabbit anti-GPR43 (Merck, 1:1000) in 1× PBS with 0.1% Tween. The mouse anti-β actin antibody (1:500, Santa Cruz, Heidelberg, Germany) was used as a housekeeping protein. After 4 °C incubation overnight, membranes were washed in Tris-buffered saline (pH 7.4) and incubated for 1 h at room temperature with secondary antibody IRDye 800CW or 680 CW (LI-COR, 1:20000) in the dark. Membranes were analyzed using a LI-COR Odyssey Scanner (LI-COR).

### 4.10. Quantitative Polymerase Chain Reaction (qPCR) 

A qPCR was performed to determine the gene expression of mitochondrial fission and fusion in human islets and MIN6 cells after acetate/butyrate and STZ incubation as described above. After stimulation, islets and MIN6 cells were homogenized with TRIzol (Life Technologies, Carlsbad, CA, USA). The total RNA was isolated following the manufacturer’s instructions, followed by obtaining reverse-transcribed cDNA using SuperScript II Reverse Transcriptase (Invitrogen). qPCR was performed with a FastStart Universal SYBR-Green Master kit (Roche) for the genes *MFN1*, *MFN2*, *DRP1*, *OPA1*, and *FIS1* (primer sequences are listed in Table 1). Reactions were carried out in 384-well PCR plates (Thermo Scientific) using a ViiA7 Real-Time PCR System (Applied Biosystems, Carlsbad, CA, USA). Delta Ct values were calculated and normalized to the housekeeping gene β-actin. Delta Delta Ct values were used for the comparative quantification of expression of the genes.

### 4.11. Statistical Analysis

Parametric distribution of data was tested using the Kolmogorov–Smirnov test. Data are expressed as mean ± standard error of mean (SEM). Statistical differences of parametric data were analyzed using one-way ANOVA, while nonparametric data were analyzed with a Kruskal–Wallis test. *p*-values < 0.05 were considered to be statistically significant (* *p* < 0.05, ** *p* < 0.01, and *** *p* < 0.001); ns, not significant. The data were analyzed using GraphPad Prism (v. 7.00; GraphPad Software Inc, La Jolla, USA).

## Figures and Tables

**Figure 1 ijms-21-01542-f001:**
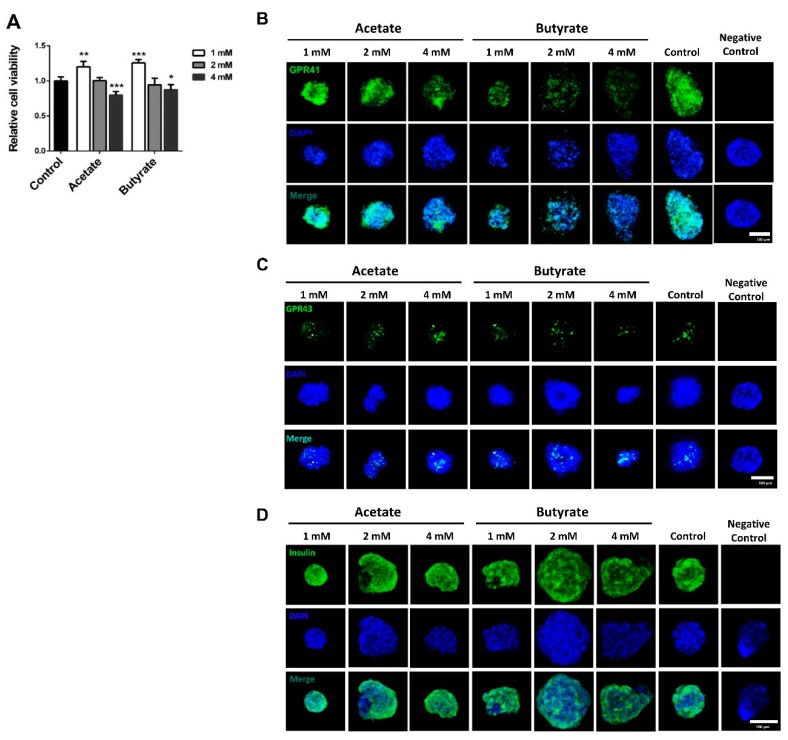
Effects of acetate and butyrate on the viability and expression of G protein-coupled receptor 41 (GPR41) and GPR43 in human islets. Human islets were incubated with 1, 2, or 4 mM acetate or butyrate for 24 h. Cell viability was quantified with water soluble tetrazolium salt 1 (WST-1) assay (**A**), and GPR41 (**B**), GRP43 (**C**), and insulin **(D**) protein were visualized with immunofluorescence. Results are plotted as mean ± SEM (*n* = 5, different donors). Islets incubated with only fluorescent-conjugated second antibody served as negative control. The statistical differences were quantified using one-way ANOVA analysis with Newman–Keuls multiple comparisons test (* *p* < 0.5, ** *p* < 0.01, *** *p* < 0.001, compared to the untreated control group). Scale bar denotes 100 µm.

**Figure 2 ijms-21-01542-f002:**
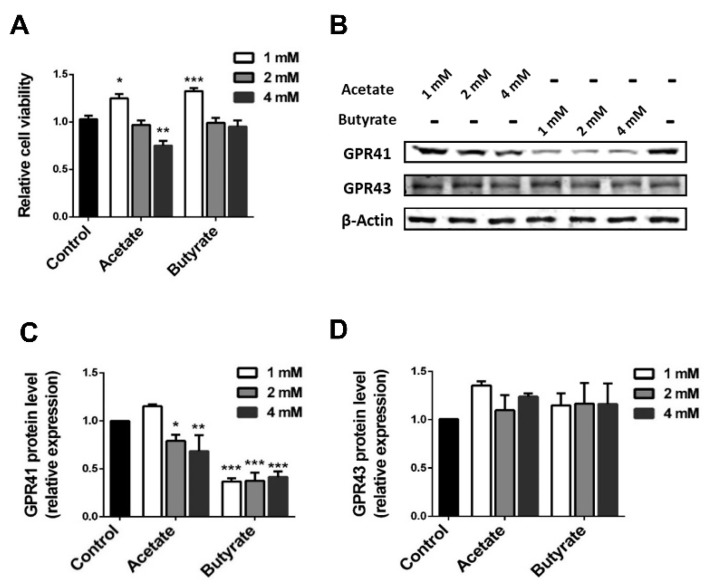
Effects of acetate and butyrate on mouse insulinoma MIN6 β-cell viability and expression of GPR41 and GPR43. MIN6 cells incubated with acetate or butyrate for 24 h. (**A**) Cell viability was determined by WST-1 assay. The expressions of GPR41 and GPR43 were investigated by Western blot (**B**). Western blot results were analyzed by using Image J gradation analysis of GRP41 (**C**) and GRP43 (**D**). Results are represented as mean ± SEM (*n* = 5). The statistical differences were quantified using one-way ANOVA analysis with Newman–Keuls multiple comparisons test (* *p* < 0.5, ** *p* < 0.01, *** *p* < 0.001, compared to the untreated control group).

**Figure 3 ijms-21-01542-f003:**
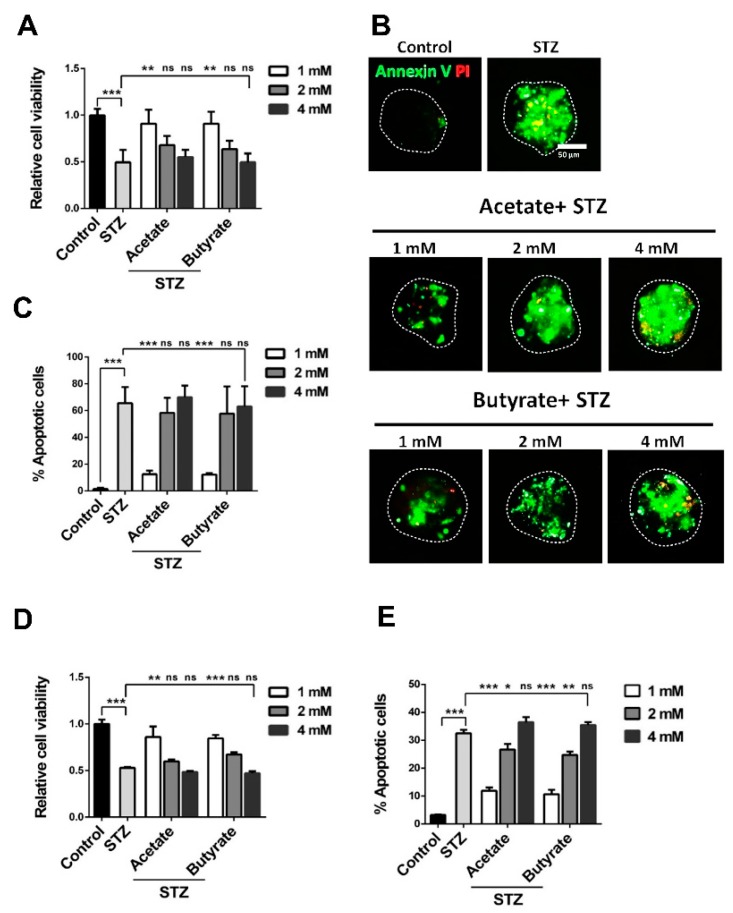
Effects of acetate and butyrate on viability and apoptosis in streptozotocin (STZ)-treated human islets and MIN6-cells. Following pre-incubation with the indicated concentration of acetate or butyrate for 1 h, islets and MIN6-cells were co-incubated with the tested short-chain fatty acids (SCFA) and 5 mM STZ for 24 h. (**A**) Islet cell viability was determined by WST-1 assay. (**B**) Islet cell apoptosis was detected by Annexin V and propidium iodide (PI) staining. Scale bar denotes 50 µm. (**C**) Islet Annexin PI staining results were analyzed by using Image J gradation analysis. Human islets were tested from 5 donors. (**D**) MIN6-cell viability was determined by WST-1 assay. (**E**) MIN6 cell apoptosis was detected using a flow cytometric assay with Annexin V and PI staining. Results are plotted as mean ± SEM (*n* = 5). The statistical differences were quantified using one-way ANOVA analysis with Newman–Keuls multiple comparisons test (* *p* < 0.5, ** *p* < 0.01, *** *p* < 0.001); ns, not significant.

**Figure 4 ijms-21-01542-f004:**
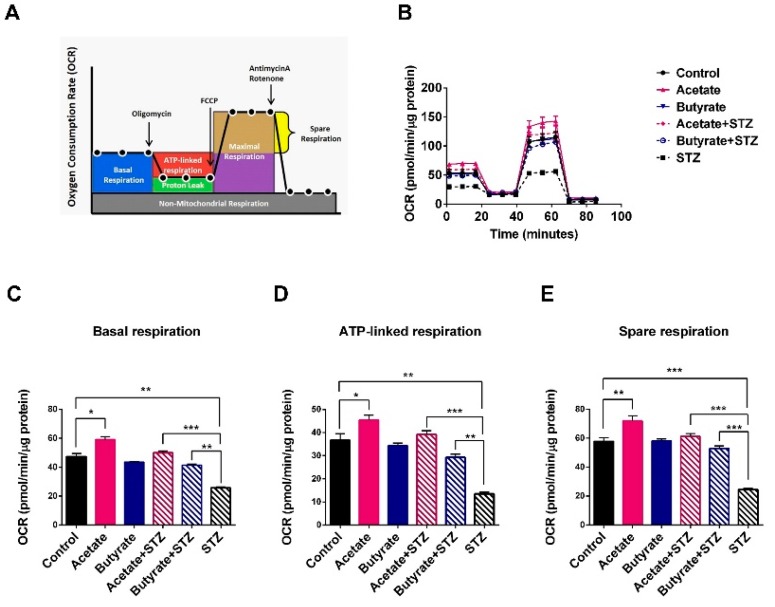
Effects of acetate and butyrate on oxygen consumption rate (OCR) of MIN6-cells. (**A**) A schematic overview of the mitochondrial stress test. Arrows indicate the subsequent addition of the ATPase inhibitor oligomycin, the uncoupling reagent phenyl hydrazone (FCCP), and the inhibitors of the electron transport chain rotenone/antimycin A. (**B**) MIN6 cell were seeded in a Seahorse cell culture plate; after incubation with 1 mM of acetate or butyrate, STZ at a final concentration of 5 mM was added to each well and incubated for 24 h. The OCR of each treated cell was investigated using a Seahorse Biosience XF24 extracellular flux analyzer. Figures **C**–**E** respectively represent individual parameters for basal respiration (**C**), ATP-linked respiration (**D**), and spare respiration (**E**). Results are plotted as mean ± SEM (*n* = 5). The statistical differences were quantified using one-way ANOVA analysis with Newman–Keuls multiple comparisons test (* *p* < 0.5, ** *p* < 0.01, *** *p* < 0.001).

**Figure 5 ijms-21-01542-f005:**
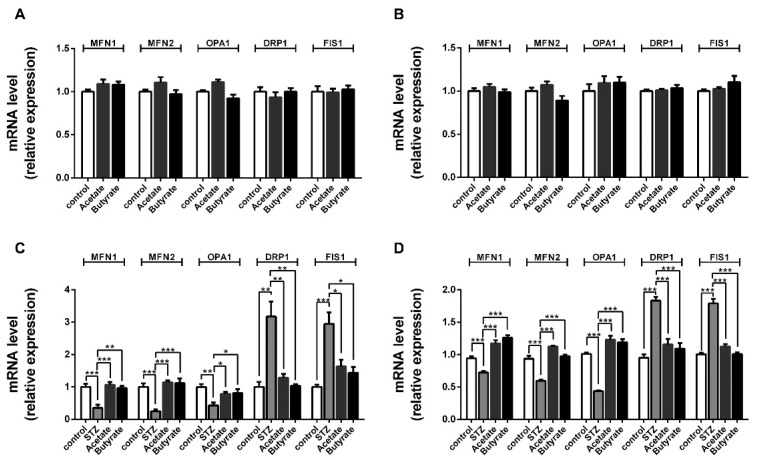
Effect of acetate and butyrate on mitochondrial fusion and fission gene expression in human islets and MIN6-cells. Islets (**A**,**C**) and MIN6-cells (**B**,**D**) were incubated with 1 mM acetate or butyrate in the absence (**A**,**B**) or presence (**C**,**D**) of 5 mM STZ for 24 h. The expression level of the mitochondrial fusion and fission genes were quantified with qPCR analysis. Results are plotted as mean ± SEM (*n* = 5). The statistical differences were quantified using one-way ANOVA analysis with Newman–Keuls multiple comparisons test (* *p <* 0.5, ** *p* < 0.01, *** *p* < 0.001).

**Figure 6 ijms-21-01542-f006:**
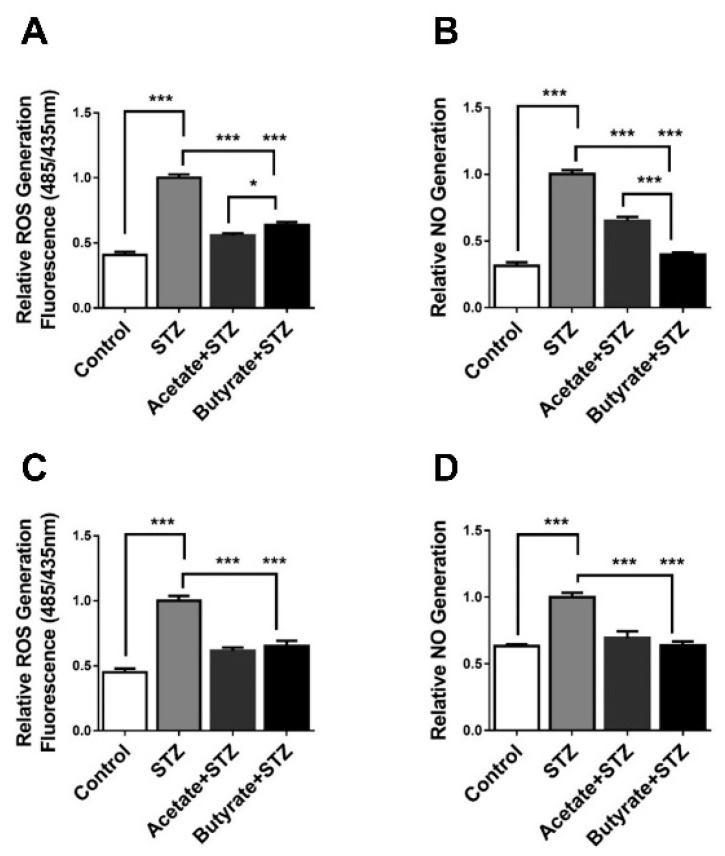
Effects of acetate and butyrate on STZ-induced free radical generation in human islets and MIN6-cells. Human islets (**A**,**B**) and MIN6-cells (**C**,**D**) were incubated with 1 mM acetate or butyrate for 1 h followed by 24 h incubation with 5 mM STZ. Intracellular reactive oxygen species (ROS) was measured with a 2′,7′-dichlorofluorescin diacetate (DCFDA) Cellular ROS Detection Assay Kit (**A**,**C**). The fluorescence signal was monitored at Ex/Em = 485/535 nm. Nitric oxide (NO) was detected using a Nitric Oxide Assay Kit (**B**,**D**). Results represent mean ± SEM (*n* = 5). The statistical differences were quantified using one-way ANOVA analysis with Newman–Keuls multiple comparisons test (* *p* < 0.5, *** *p* < 0.001).

**Table 1 ijms-21-01542-t001:** Primer sequences for qRT-PCR.

Gene	Species	Forward Sequence 5′-3′	Reverse Sequence 5′-3′
*MFN1*	Mouse	TCTCCAAGCCCAACATCTTCA	ACTCCGGCTCCGAAGCA
*MFN2*	Mouse	ACAGCCTCAGCCGACAGCAT	TGCCGAAGGAGCAGACCTT
*DRP1*	Mouse	GCGCTGATCCCGCGTCAT	CCGCACCCACTGTGTTGA
*OPA1*	Mouse	TGGGCTGCAGAGGATGGT	CCTGATGTCACGGTGTTGATG
*FIS1*	Mouse	GCCCCTGCTACTGGACCAT	CCCTGAAAGCCTCACACTAAGG
*β-Actin*	Mouse	ACGGCCAGGTCATCACTATTC	AGGAAGGCTGGAAAAGAGCC
*MFN1*	Human	TGGCTAAGAAGGCGATTACTGC	TCTCCGAGATAGCACCTCACC
*MFN2*	Human	CTCTCGATGCAACTCTATCGTC	TCCTGTACGTGTCTTCAAGGAA
*DRP1*	Human	CTGCCTCAAATCGTCGTAGTG	GAGGTCTCCGGGTGACAATTC
*OPA1*	Human	TGTGAGGTCTGCCAGTCTTTA	TGTCCTTAATTGGGGTCGTTG
*FIS1*	Human	GTCCAAGAGCACGCAGTTTG	ATGCCTTTACGGATGTCATCATT
*β-Actin*	Human	GCACCACACCTTCTACAATG	TGCTTGCTGATCCACATCTG

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
