# Peer review of "Acetate and Butyrate Improve β-cell Metabolism and Mitochondrial Respiration under Oxidative Stress"

_ijms, 2020, doi:10.3390/ijms21041542_

Round 1

Reviewer 1 Report

In this manuscript entitled "Acetate and butyrate improve β-cell metabolism and mitochondrial respiration under oxidative stress", Hu et al., elegantly present the effects of SCFAs on pancreatic beta cell functions with regards to its viability under stress conditions. The paper is well designed and clearly written. There a few minor issues that the authors may want to consider to improvise this manuscript.

General minor comments:

The authors have not done any insulin staining of their islets. A panel of insulin stainings alongside in Fig. 1B with the SCFA exposure will be more informative. Insulin staining is important to make sure these are "islets".

Following up on the above comment, the islets look a bit weird. Why did the authors not use confocal microscopy to visualize the islets? The authors' statement on the reduction of GPR41 (line# 78) cannot be considered as they did not perform confocal imaging.

Line #251 spell check "fluorescence sinal" -- signal

Author Response

Point 1: The authors have not done any insulin staining of their islets. A panel of insulin stainings alongside in Fig. 1B with the SCFA exposure will be more informative. Insulin staining is important to make sure these are "islets".

Response 1: In the study, we focus on β-cell survival exposed to SCFAs under stress. Our result shows significant improvement after a pre-incubation with SCFAs, which indicates an increased survival rate of insulin-producing β-cell. As suggested by the reviewer we have stained the islets for insulin and included illustrations on page 3 line 85 but we did not see any effect on insulin staining of the SCFA. This result is now mentioned on page 2 lines 82-84. Details of the antibody staining is mentioned on page 12 lines 431 and 433 of the materials and methods section.

Point 2: Following up on the above comment, the islets look a bit weird. Why did the authors not use confocal microscopy to visualize the islets? The authors' statement on the reduction of GPR41 (line# 78) cannot be considered as they did not perform confocal imaging.

Response 2: We used the hanging droplet methodology as we had too many samples for confocal. With this method we can study the islets at different depths. This is now mentioned on page 12 lines 435-438. We included a higher quality picture in Figure 1B. The figure is inserted in our revised manuscript on page 3 line 85. We hope the new figure more clearly demonstrate the pertinent difference in GRP41 expression between the different groups.

Point 3: Line #251 spell check "fluorescence sinal" – signal

Response 3: We apologize for this spelling mistake.  In line#257 "sinal" has been changed to "signal".

Reviewer 2 Report

The study describes the protective effect of SCFAs on beta-cells under oxidative stress.  The study is technically weak. The results are not conclusive to believe SCFAs effect on beta cells. I suggest the following revision for the study.

Oxidative stress mostly induce islet or beta-cell dysfunction. The study requires static incubation or other functional assays to check the SCFAs effect on glucose-mediated response. The study mainly focuses on SCFAs protective role on Beta cells. Primary beta cells sorted from human islets will be an appropriate tool for this study. Min6 is a genetically modified cell line model, its function and signaling mechanism greatly differ from primary beta cells.

Include Islet or primary sorted beta cell-based OCR experiment results to prove SCFAs improve beta-cell specific oxidative or metabolic stress. Again, Min-6 OCR experiments are not convincing for this study. Otherwise in the title, instead of beta cells, use the term min-6 cells.

There is no novelty in the study. The potential role of Short-chain fatty acids (SCFAs) on beta cells or islets are well reported already (Pingitore A et al., 2017 & 2019). Discuss your results with these similar studies.

Figure 3, authors had mentioned n=5. This are biological or experimental repeats? Whether these islets came from the same or different pancreas? How the STZ dose was optimized. What is the magic of using 5mM of STZ to induce oxidative and metabolic stress in both islets and min6 cell lines? These cells differ greatly in proliferation and other signaling mechanisms.

Author Response

Point 1: Oxidative stress mostly induce islet or beta-cell dysfunction. The study requires static incubation or other functional assays to check the SCFAs effect on glucose-mediated response. The study mainly focuses on SCFAs protective role on Beta cells. Primary beta cells sorted from human islets will be an appropriate tool for this study. Min6 is a genetically modified cell line model, its function and signaling mechanism greatly differ from primary beta cells.

Response 1: We used OCR and not static incubation as it has been reported that OCR correlates much better and shows smaller differences in viability and function than static incubation [1]. This is now mentioned more clearly on page 6 lines 162-164. For the inclusion of human islets we feel there is a misunderstanding. Both human islets and the MIN6 β-cell line were used in the study to show the impact of SCFAs on real islets and insulin-producing β-cell. We investigated human islet viability, apoptosis, expression of GPR41 and GPR43, mitochondrial dynamics and free radical release. Results are shown in Figures 1, 3, 5 and 6. ‘Primary β-cells sorted from human islets’ is basically impossible as it requires too many islets as isolating β-cells is associated with massive loss of cells and impaired function. We have done this many times in previous studies but it was impossible to do this for the current studies as it requires too many donor islets. We therefore choose the strategy to use MIN6 cells for the assays where high numbers of islets are needed. This is now mentioned more clearly on page 9 lines 276-281 of the discussion section.

Point 2: Include Islet or primary sorted beta cell-based OCR experiment results to prove SCFAs improve beta-cell specific oxidative or metabolic stress. Again, Min-6 OCR experiments are not convincing for this study. Otherwise in the title, instead of beta cells, use the term min-6 cells.

Response 2: As mentioned in response to the previous comment, it is impossible to do many of the assays with sorted β-cells isolated from human islets. The OCR assay could only be performed with MIN6 cells and not with rarely available human islets. However, we feel we have enough convincing scientifically sound novel data to demonstrate efficacy of the SCFAs on both human islets and the MIN6 cell line. E.g. we demonstrate a pro-fusion and anti-fission effect of the tested SCFAs on islets compromised by streptozotocin, which also indicates a diminished mitochondrial dysfunction under stress.

Point 3: There is no novelty in the study. The potential role of Short-chain fatty acids (SCFAs) on beta cells or islets are well reported already (Pingitore A et al., 2017 & 2019). Discuss your results with these similar studies.

Response 3: Pingitore A et al. investigated the effect of acetate and propionate on β-cell function compromised by cytokines and sodium palmitate to investigate effects on potentiation of insulin secretion. Our study focuses on the impact of SCFAs on islet β-cell survival under stress with another focus as it may delay diabetes onset. We went into mitochondrial effects and demonstrate clear differences between efficacy between SCFAs which is to our opinion novel. Also, as far as we can oversee butyrate was not investigated which shows in our study promising effects on supporting and rescuing β-cell under homeostasis and stressful conditions. Also we studied impact after exposure to STZ, a more specific toxin to β-cell. Additionally, we demonstrate that both acetate and butyrate rescue STZ-impaired β-cells mitochondrial respiration and promote mitochondrial repair by promoting a shift toward the fusion process. This is, to the best of our knowledge, a new mechanism by which SCFAs contribute to rescue of damaged β-cells. As requested by the reviewer we discussed the differences between the studies of Pingitore A et al and ours more clearly on page 10 lines 317-330 of the discussion section.

Point 4: Figure 3, authors had mentioned n=5. This are biological or experimental repeats? Whether these islets came from the same or different pancreas? How the STZ dose was optimized. What is the magic of using 5mM of STZ to induce oxidative and metabolic stress in both islets and min6 cell lines? These cells differ greatly in proliferation and other signaling mechanisms.

Response 4: All experiments in the study were performed with 5 different donors. It is mentioned under Figure 3 in Line#136 as “Human islets were tested from 5 donors”. All the islet and donor information are present in the supplementary information Table A. For the concentration of STZ, we did a pilot incubating MIN6 β-cell and human islet with STZ at 1, 2, and 5 mM. Only 5 mM STZ induce significant viability decrease in both MIN6 β-cell and human islets.

  1. Papas, K.K.; Bellin, M.D.; Sutherland, D.E.R.; Suszynski, T.M.; Kitzmann, J.P.; Avgoustiniatos, E.S.; Gruessner, A.C.; Mueller, K.R.; Beilman, G.J.; Balamurugan, A.N., et al. Islet Oxygen Consumption Rate (OCR) Dose Predicts Insulin Independence in Clinical Islet Autotransplantation. PLOS ONE 2015, 10, e0134428, doi:10.1371/journal.pone.0134428.

Round 2

Reviewer 2 Report

The authors have answered all the queries. Thanks